# Probiotic Incorporation into Yogurt and Various Novel Yogurt-Based Products

Douglas W. Olson and Kayanush J. Aryana *

School of Nutrition and Food Sciences, Dairy Science Building, Louisiana State University Agricultural Center, Baton Rouge, LA 70803, USA
* Correspondence: karyana@agcenter.lsu.edu; Tel.: +1-225-578-4380

**Abstract:** Probiotics are commonly added to yogurt to provide many health benefits for the consumer. A description is provided for some commonly used probiotics in yogurt. A GRAS (generally recognized as safe) list of probiotic bacteria that can be added to yogurt or similar types of products is provided. Additionally, prebiotics, synbiotics (combination of prebiotics and probiotics), postbiotics, paraprobiotics, and psychobiotics can be added to yogurt. Probiotic yogurt can come in various forms in addition to spoonable yogurt, and yogurt can be used as an ingredient in other food products. Many useful functional ingredients can be applied to probiotic yogurt. The safety of probiotics must be addressed, especially for critically ill patients and other susceptible populations. Probiotics must survive within yogurt throughout its entire shelf-life and within the gastrointestinal tract after consumption by the consumer to provide health benefits, and many techniques can be used to maintain survival of probiotics in yogurt. Furthermore, probiotics can be added to Greek yogurt acid whey. Many opportunities exist for adding a wide variety of probiotics to a wide variety of yogurt-based products.

**Keywords:** probiotic; fermented; yogurt; health

## 1. Introduction

Legal requirements of the U.S. Food and Drug Administration for yogurt are stated in 21 CFR 131.200 (www.ecfr.gov) (accessed on 24 August 2022) [1]. Yogurt is basically described as the food produced by culturing certain types of dairy ingredients with a bacterial culture that includes *Lactobacillus delbrueckii* subsp. *bulgaricus* and *Streptococcus thermophilus*. Optional ingredients that may be added to yogurt include other cultures, nutritive carbohydrate sweeteners, flavoring ingredients, color additives, stabilizers, emulsifiers, preservatives, and vitamins A and D. Dairy ingredients must be pasteurized or ultra-pasteurized and may be homogenized before addition of the culture. Before addition of bulky flavorings, yogurt must contain at least 3.25% milkfat and 8.25% milk solids-not-fat and either have a titratable acidity of at least 0.7% (expressed as lactic acid) or a pH of 4.6 or less. The phrase "contains live and active cultures" may be displayed on the package if there are at least $10^7$ cfu/g of viable bacteria when manufactured and if it can reasonably be expected to have $10^6$ cfu/g during its entire shelf-life. The viable microorganisms may be inactivated after culturing to prolong the shelf-life of yogurt. The definitions and standards of identity for low-fat yogurt (previously described in 21 CFR 131.203) and non-fat yogurt (previously described in 21 CFR 131.206) have been revoked as of 7 July 2021 (www.federalregister.gov) (accessed on 24 August 2022) [2].

Sales of yogurt and probiotics are large and still growing. According to a 2021 report by Statista, U.S. sales of yogurt were $7.24 billion in 2021 compared to $5.58 billion in 2011 [3]. The Greek yogurt share of the yogurt market was 51% in 2021 [4]. Frozen yogurt production in the U.S. was 46.4 million gallons in 2021 [5]. The annual growth of probiotic yogurt was 6.6% in the U.S. in 2020 compared to 11% for the overall yogurt market [6].

The global value of the probiotics market was $58 billion in 2021 and is predicted to grow at an annual rate of 7.5% until 2030 [7], and the global probiotic drink market was worth $13.65 billion in 2019 with an expected annual growth rate of 6.1% from 2020 to 2027 [8].

## 2. History of Discovery and Definitions of Probiotics

Experiments for studying effects of bacteria on treating health problems and promoting good health have been performed for a long time. Theodor Escherich has been credited as the first pediatric infectious disease physician and described *Bacterium coli commune* (now referred to as *Escherichia coli*) in 1886 [9]. While working under Theodor Escherich, Dr. Józef Brudziński treated infants for acute infectious diarrhea by using a *Bacillus lactis aërogenes* suspension described in publications from 1899 [10,11]. Although Élie Metchnikoff [12] believed that intestinal putrefaction can shorten life, he noted the work of Dr. Brudziński and similar work by Dr. Henry Tissier and recommended people "to absorb large quantities of microbes". He believed that lactic bacteria can fight against intestinal putrefaction. He also wrote that Stamen Grigoroff observed many centenarians in Bulgaria, which is a region where yahourth (yogurt) was commonly consumed [12]. The fact that diet affects the types of bacteria that develops within the intestinal tract was first clearly established by Herter and Kendall in 1910, but suggested as early as 1886 by Escherich and Hirschler [13].

Many of the starter cultures and probiotics now used in yogurt making were first described in the late 1800s or early 1900s. The name "*Streptococcos*" was first used in 1874 by Albert Theodor Billroth [14]. *Streptococcus thermophilus* (later reclassified as *Streptococcus salivarius* subsp. *thermophilus* by Farrow and Collins in 1984 [15] but revived back to *Streptococcus thermophilus* by Schleifer et al. in 1991 [16]) was described by S. Orla-Jensen in 1919 [17]. In 1901, Martinus Beijerinck proposed the genus *Lactobacillus* to include Gram-positive, fermentative, facultatively anaerobic, non-sporeforming bacteria [18]. Stamen Grigoroff discovered Bulgarian bacillus (now *Lactobacillus delbrueckii* ssp. *bulgaricus*) in 1905 [19]. *Lactobacillus acidophilus* (originally called *Bacillus acidophilus*) was described by Ernst Moro in 1900 [20]. In 1899 and 1900, Henry Tissier first described *Bacillus bifidus communis*, later referred to as *Lactobacillus bifidus* and now referred to as *Bifidobacterium* [21]. He found that *Bifidobacteria* was the main type of bacteria comprising the gut microflora of breast-fed babies and *Bifidobacteria* could treat acute gastroenteritis [19].

Dr. Isaac Carosso recommended to his patients who suffered from gastrointestinal problems to consume yogurt. Afterwards, he started producing yogurt and founded the Danone Company in 1919 [19].

The term "probiotic" (meaning "for life") originated in 1953 from Werner Kollath to mean "active substances that are essential for a healthy development of life" [22]. Lilly and Stillwell [23] used the term probiotic as "substances secreted by one organism which stimulate the growth of another" in 1965. Parker [24] described probiotics as "organisms and substances which contribute to intestinal microbial balance" in 1974. Fuller [25] defined probiotics as "A live microbial feed supplement which beneficially affects the host animal by improving its intestinal microbial balance" in 1989. A panel from the International Scientific Association for Probiotics and Prebiotics defined probiotic as "live microorganisms that, when administered in adequate amounts, confer a health benefit on the host" in 2014 [26].

## 3. Criteria for and Types of Probiotics and Its Use in Foods

In order for a microorganism to be classified as a probiotic, the microorganism must be properly characterized, safe for its intended use, shown to confer a health benefit to the host by at least one human clinical trial, and be viable at the required dose until the end of the product shelf life [27]. In general, probiotic viability is affected by temperature, water activity, pH, ingredients, oxygen level, packaging materials, and storage time [28]. Selection criteria for incorporation of a probiotic into a food includes being of a human or food origin, safe for human consumption, maintaining desirable properties in the final product, survival during passage through the gastrointestinal tract including sufficient acid and bile tolerance, adhesion to intestinal mucosal surface, and providing proven health benefits to the host [29]. For example, survival of greater than 7 log cfu/g for

*Limosilactobacillus* (formerly *Lactobacillus*) *mucosae* CNPC007 that was incorporated into goat milk Greek-style yogurt was found after exposure to simulated digestion, allowing this strain to be considered a possible probiotic [30].

Probiotics that confer health benefits or pathogens that cause diseases are often strain specific. In some cases, a certain strain of a species may be a probiotic while another strain of the same species may be a pathogen. For example, *E. coli* Nissle 1917 is a probiotic [31], while *E. coli* O157:H7 and O104:H4 are pathogens. *Bacillus* sp. DU-106 from the *Bacillus cereus* group is a potential probiotic but other strains of *B. cereus* are pathogens [32]. Some strains of *Clostridium butyricum* can be used as a probiotic to produce bacteriocins, secrete butyrate, and inhibit pathogens while other strains are linked with botulism in infants and necrotizing enterocolitis in preterm neonates [33]. Sometimes, probiotics can cause illness under certain conditions, so potential safety issues must be addressed as will be discussed in more detail later.

There are many types of probiotic bacteria. Bacteria and yeast with claimed probiotic or potential probiotic properties that have been incorporated into or isolated from yogurt or yogurt-related products are listed in Table 1 [34–72]. A list of GRAS (generally recognized as safe) probiotics that mention incorporation into yogurt or related milk products is provided in Table 2. GRAS rules originated from the Food Additives Amendment of 1958 to the Federal Food, Drug, and Cosmetic Act of 1938 (sections 201(s) and 409) and require successful safety evaluations for their intended use by experts in the field, unless it has been shown to be safe by common use before 1958 [73]. For example, Endres et al. [74] demonstrated that *Bacillus coagulans* GBI-30, 6086 is safe for human consumption.

**Table 1.** Bacteria, including postbiotics, and yeast with claimed probiotic properties, potential probiotic properties, or potential therapeutic application that have been incorporated into or isolated from yogurt or yogurt-related products.

| Bacteria | Reference |
|---|---|
| *Lactobacillus delbrueckii* ssp. *bulgaricus* | All yogurt |
| *Lactobacillus delbrueckii* ssp. *lactis* | [34] |
| *Lactobacillus acidophilus* | [35] |
| *Lactobacillus amylovorus* | [36] |
| *Lactobacillus crispatus* | [37] |
| *Lactobacillus gasseri* | [38] |
| *Lactobacillus helveticus* | [39] |
| *Lactobacillus jensenii* | [40] |
| *Lactobacillus johnsonii* | [37] |
| *Lacticaseibacillus casei* (formerly *Lactobacillus casei*) | [41] |
| *Lacticaseibacillus paracasei* (formerly *Lactobacillus casei* or *Lactobacillus paracasei*) | [42] |
| *Lacticaseibacillus rhamnosus* (formerly *Lactobacillus rhamnosus*) | [43] |
| *Lactiplantibacillus plantarum* (formerly *Lactobacillus plantarum*) | [44] |
| *Lactiplantibacillus paraplantarum* (formerly *Lactobacillus paraplantarum*) | [45] |
| *Lactiplantibacillus pentosus* (formerly *Lactobacillus pentosus*) | [46] |
| *Latilactobacillus sakei* (formerly *Lactobacillus sakei*) | [47] |
| *Latilactobacillus curvatus* (formerly *Lactobacillus curvatus*) | [48] |
| *Lentilactobacillus parafarraginis* (formerly *Lactobacillus parafarraginis*) | [49] |
| *Levilactobacillus brevis* (formerly *Lactobacillus brevis*) | [50] |
| *Ligilactobacillus salivarius* (formerly *Lactobacillus salivarius*) | [51] [1] |
| *Limosilactobacillus fermentum* (formerly *Lactobacillus fermentum*) | [36] |
| *Limosilactobacillus mucosae* (formerly *Lactobacillus mucosae*) | [30] |
| *Limosilactobacillus reuteri* (formerly *Lactobacillus reuteri*) | [52] |
| *Loigolactobacillus coryniformis* (formerly *Lactobacillus coryniformis*) | [38] |
| *Weisella* (formerly *Lactobacillus*) *viridescens* | [34] |
| *Weisella cibaria* | [53] |
| *Weisella paramesenteroides* | [45] |
| *Streptococcus thermophilus* | All yogurt |
| *Streptococcus salivarius* | [54] |

**Table 1.** *Cont.*

| Bacteria | Reference |
|---|---|
| *Bifidobacterium bifidum* | [55] |
| *Bifidobacterium longum* | [55] |
| *Bifidobacterium infantis* | [55] |
| *Bifidobacterium adolescentis* | [55] |
| *Bifidobacterium breve* | [55] |
| *Bifidobacterium animalis* ssp. *lactis* | [56] |
| *Weizmannia coagulans* (formerly *Bacillus coagulans*) | [57] |
| *Bacillus subtilis* | [58] |
| *Priestia flexa* (formerly *Bacillus flexus*) | [58] |
| *Bacillus licheniformis* | [58] |
| *Bacillus mojavensis* | [58] |
| *Bacillus amyloliquefaciens* [2] | [59] |
| *Propionibacterium freudenreichii* ssp. *shermanii* | [60] |
| *Propionibacterium acidipropionici* | [61] |
| *Propionibacterium jensenii* | [62] |
| *Propionibacterium thoenii* (*jensenii*) | [62] |
| *Leuconostoc mesenteroides* | [63] |
| *Leuconostoc pseudomesenteroides* | [64] |
| *Leuconostoc lactis* | [65] |
| *Lactococcus lactis* ssp. *lactis* (formerly *Streptococcus lactis*) | [44] |
| *Lactococcus cremoris* (formerly *Streptococcus cremoris*) | [66] [3] |
| *Pediococcus acidilactici* | [67] |
| *Pediococcus pentosaceus* | [68] |
| *Enterococcus faecium* | [69] |
| *Enterococcus faecalis* | [45] |
| *Enterococcus durans* | [69] |
| *Enterococcus lactis* | [68] |
| *Bacteroides vulgatus* | [70] [4] |
| *Bacteroides dorei* | [70] [4] |
| *Faecalibacterium prausnitzii* | [70] [4] |
| *Prevotella copri* | [70] [4,5] |
| **Yeast** | |
| *Saccharomyces cerevisiae var. boulardii* | [71] |
| *Kluyveromyces marxianus* | [72] |

[1] This reference describes adding this probiotic within jelly candy enriched with grape seeds extract. [2] This probiotic was isolated from yogurt-flavored cultured beverage Yogu Farm™. [3] Although this reference stated that *Lactococcus lactis* ssp. *cremoris* was isolated from yogurt, other papers have described this bacterium as a probiotic. [4] These new generation probiotics were found in homemade back-slopped yogurts. [5] Although this bacterium has been associated with health benefits, an overabundance of this intestinal bacterium was associated with arthritis and intestinal mucositis.

**Table 2.** List of GRAS (generally recognized as safe) substances (viable probiotic, heat-killed microorganism, or spore preparation) that might be able to be used in yogurt or related products as of 30 October 2022. Taken from: https://www.cfsanappsexternal.fda.gov/scripts/fdcc/?set=GRASNotices (accessed on 30 October 2022).

| GRN Number | Substance (Beneficial Microorganism) | Notifier | Status [1] | Date of Closure |
|---|---|---|---|---|
| 1065 | *Anaerobutyricum soehngenii* CH106 | Caelus Health | Pending | |
| 1063 | *Weissella cibaria* CMU | OraPharm, Inc. | Pending | |
| 1022 | *Streptococcus salivarius* DB-B5 | Dose Biosystems | No questions | 8/22/2022 |
| 1013 | *Lactobacillus rhamnosus* DSM 33156 (LGG) | Chr. Hansen's, Inc. | No questions | 12/15/2022 |
| 1003 | *Bifidobacterium longum* subsp. *infantis* M-63 | Morinaga Milk Industry Co., Ltd. | No questions | 4/26/2022 |
| 1002 | *Bifidobacterium breve* strain MCC1274 | Morinaga Milk Industry Co., Ltd. | No questions | 7/22/2022 |
| 988 | *Lactobacillus fermentum* LfQi6 | Quorum Innovations | No questions | 3/28/2022 |

**Table 2.** *Cont.*

| GRN Number | Substance (Beneficial Microorganism) | Notifier | Status [1] | Date of Closure |
|---|---|---|---|---|
| 971 | *Bacillus clausii* 088AE spore preparation | Advanced Enzyme Technologies | No questions | 3/3/2022 |
| 969 | *Bacillus subtilis* "Bss-19" spore preparation | Danisco USA | No questions | 10/6/2021 |
| 957 | *Lactobacillus johnsonii* strain ATCC PTA-124205 | Prozure, Inc. | No questions | 10/26/2021 |
| 956 | *Bacillus subtilis* ATCC SD-7280 | Advanced Enzyme Technologies | No questions | 8/18/2021 |
| 955 | *Bacillus subtilis* strain BS-MB40 PTA-122264 spore preparation | BIO-CAT Microbials | No questions | 3/26/2021 |
| 953 | *Lactobacillus plantarum* strain CECT 7527, CECT 7528, and CECT 7529 | Kaneka Americas Holding | No questions | 2/5/2021 |
| 952 | *Bifidobacterium animalis* subsp. *lactis* strain AD011 | BIFIDO CO., LTD. | No questions | 3/17/2021 |
| 950 | *Bifidobacterium longum* subsp. *infantis* DSM 33361 | Chr. Hansen | No questions | 3/1/2021 |
| 949 | *Bacillus coagulans* strain DSM 17654 spore preparation | Advanced Enzyme Technologies Ltd. | No questions | 1/7/2021 |
| 875 | *Bifidobacterium animalis* subsp. *lactis* AD011 | BIFIDO CO., LTD. | No questions | 10/30/2019 |
| 872 | *Bifidobacterium animalis* subsp. *lactis* UAB1a-12 | UAS Laboratories | No questions | 12/9/2019 |
| 871 | *Lactobacillus acidophilus* DDS-1 | UAS Laboratories | No questions | 10/23/2019 |
| 856 | *Bifidobacterium animalis* subsp. *lactis* strain BB-12 | Chr. Hansen | No questions | 12/9/2019 |
| 847 | *Lactobacillus plantarum* ECGC 13110402 | ProBiotix Health | No questions | 9/30/2019 |
| 845 | *Lactobacillus rhamnosus* GG | Chr. Hansen | No questions | 10/30/2019 |
| 831 | *Bacillus subtilis* DE111 | Deerland Probiotics | No questions | 8/13/2019 |
| 820 | *Lactobacillus fermentum* CECT 5716 | Biosearch. S.A. | No questions | 4/3/2019 |
| 814 | *Bifidobacterium bifidum* BGN4 | BIFIDO Co., Ltd. | No questions | 6/25/2019 |
| 813 | *Bifidobacterium longum* BORI | BIFIDO Co., Ltd. | No questions | 6/21/2019 |
| 807 | *Streptococcus salivarius* M18 | BLIS Technologies | No questions | 6/6/2019 |
| 736 | *Lactobacillus casei* subsp. *paracasei* Lpc-37 | Du Pont Nutrition and Health | No questions | 4/11/2018 |
| 722 | *Lactobacillus plantarum* Lp-115 | Du Pont Nutrition and Health | No questions | 2/16/2018 |
| 691 | *Bacillus coagulans* SANK 70258 spore preparation | Mitsubishi-Kagaku Foods Corporation | No questions | 8/28/2017 |
| 685 | *Lactobacillus plantarum* strain 299v | Probi AB | No questions | 10/31/2017 |
| 670 | Inactivated *Bacillus coagulans* GBI-30, 6086 | Ganeden, Inc. | No questions | 3/15/2017 |
| 601 | *Bacillus coagulans* SBC 37-01 spore preparation | Sabinsa Corp. | No questions | 4/28/2016 |
| 597 | *Bacillus coagulans* SNZ 1969 spore preparation | Sanzyme Limited | No questions | 2/29/2016 |
| 591 | *Streptococcus salivarius* K12 | BLIS Technologies Ltd. | No questions | 1/25/2016 |
| 526 | *Bacillus coagulans* strain Unique IS2 spores preparation | Unique Biotech Limited | No questions | 3/23/2015 |
| 502 | *Lactobacillus acidophilus* La-14 | Danisco USA, Inc. | No questions | 8/19/2014 |
| 453 | *Bifidobacterium breve* M-16V | Morinaga Milk Industry Co., Inc. | No questions | 9/27/2013 |
| 445 | *Bifidobacterium animalis* subsp. *lactis* strains HN019, Bi-07, B1-04, and B420 | Danisco USA, Inc. | No questions | 4/10/2013 |
| 440 | *Lactobacillus reuteri* strain NCIMB 30242 | Micropharma Ltd. | No questions | 2/12/2013 |
| 429 | *Lactobacillus casei* strain Shirota | Yakult Honsha Co., Ltd. | No questions | 12/10/2012 |
| 415 | Heat-killed *Propionibacterium freudenreichii* ET-3 culture (powder) | Meiju Co., Ltd. | No questions [2] | 12/26/2012 |
| 399 | Preparation of *Bacillus coagulans* strain GBI-30, 6086 spores | Ganeden, Inc. | No questions | 7/31/2012 |

| GRN Number | Substance (Beneficial Microorganism) | Notifier | Status [1] | Date of Closure |
|---|---|---|---|---|
| 377 | *Bifidobacterium animalis* subsp. *lactis* strain Bf-6 | Cargill, Inc. | No questions | 9/29/2011 |
| 357 | *Lactobacillus acidophilus* NCFM | Danisco USA, Inc. | No questions | 4/19/2011 |
| 288 | *Lactobacillus rhamnosus* strain HN001 | Fonterra Co-operative Group | No questions | 11/1/2009 |
| 268 | *Bifidobacterium longum* strain BB536 | Morinaga Milk Industry Co., Ltd. | No questions | 7/8/2009 |
| 254 | *Lactobacillus reuteri* strain DSM 17938 | BioGaia AB | No questions | 5/29/2008 |

[1] "No questions" means "FDA has no questions". [2] Some uses may require a color additive petition.

The most common probiotics (other than the starter cultures) that are found in yogurt include species from the former *Lactobacillus* genus, the *Bifidobacterium* genus, and the former *Bacillus* genus. Recently, the *Lactobacillus* genus has been divided into 26 lineages with 23 novel genera [14] and these novel genera related to dairy foods have been reviewed by Oberg et al. [75]. A list of *Bifidobacterium* species can be found in the taxonomy browser [76]. The genus *Bacillus* has recently been reclassified as to only consisting of *B. subtilis* and *B. cereus* [77]. *Bacillus coagulans* has been renamed *Weizmannia coagulans*.

Next generation probiotics are potentially beneficial bacteria that are newly identified, non-conventional, and native to the gut microbiota and have possible therapeutic properties. *Akkermansia muciniphila*, *Bacteroides* species, certain *Bifidobacterium* species, *Christensenella minuta*, certain *Clostridium* species, *Eggerthellaceae* family, certain *Enterococcus* species, *Faecalibacterium prausnitzii*, certain lactic acid bacteria, *Parabacteroides goldsteinii*, *Pediococcus pentosaceus*, *Prevotella copri*, and certain *Streptococcus* species including *S. dentisani* 7746 and 7747 are possible next generation probiotics [78]. *Enterococcus mundtii* QAUEM2808 was isolated from dahi (an artisanal fermented milk product) and has potential to be used as an adjunct culture for fermenting milk [79]. *Weissella paramesenteroides* MYPS5.1 is another potential probiotic strain that has been isolated from a dairy source [80]. *Oscillospira* could be developed as a next generation probiotic because of beneficial microbial traits and have future applications in food, nutraceuticals, and biopharmaceuticals [81].

Also, certain probiotics can be bioengineered. *Escherichia coli* Nissle 1917 can be metabolically engineered to enhance production of heparosan, which is an acidic polysaccharide used in heparin biosynthesis and drug delivery [82]. Further examples of bioengineered probiotics prepared for useful purposes are described below.

Probiotics can be used in a wide variety of human foods [83] and in animal nutrition and health [84]. In addition to cow milk, milk from goats [85], sheep [86], buffaloes [87], yaks [88], camels [89], horses [90], and donkeys [91] has been used to produce probiotic yogurt. Dairy sources constitute 80% of the more than 380 types of probiotic products available worldwide [92]. In addition to yogurt, some types of dairy-based foods that have incorporated probiotics within research studies include milk [93], infant formula [94], kefir [95], buttermilk [96], butter [96], sour cream [97], ice cream [98], cottage cheese [93], white pickled cheese [99], Cheddar cheese [100], and Mozzarella cheese [101]. The non-dairy products include various types of soy-based yogurt [102], wheat germ [103], dehydrated wheat/rice cereal matrices [104], fruit and vegetable matrices [105], fruit and vegetable juices [8], unfiltered and unpasteurized beer [106], coffee brews [107], fermented meat products [108], chocolate [109], non-fat whipping cream analogues [110], and a milk and maize African beverage [111]. *Bacillus* spores can be used in baking due to their high heat resistance. Permpoonpattana et al. [112] found just over a 1-log reduction in viability of *Bacillus subtilis* HU58 and PXN21 lyophilized spores after baking wholemeal biscuits at 235 °C for 8 min.

## 4. Gut Microbiome, Inflammation, and Health Benefits Provided by Probiotics

The human gut microbiome (also known as microbiota or microflora) consists of bacteria (predominantly obligate anaerobes), archaea, fungi, and protists and functions

by metabolizing nutrients (by converting indigestible carbohydrates into short-chain fatty acids) for the host, maintaining the gut mucosal barrier, modifying the immune system, inhibiting pathogens, and even affecting brain activities. Most of these bacteria belong to the Firmicutes and Bacteroidetes phyla with fewer bacteria belonging to Actinobacteria, Proteobacteria, Fusobacteria, and Verrucomicrobia phyla. Firmicutes bacteria are Gram-positive and are involved in short chain fatty acid synthesis and in hunger and satiety regulation [113]. Bacteroidetes bacteria are Gram-negative and are involved with enhancing immune reactions and inflammation. A loss of a balanced ratio between Firmicutes and Bacteroidetes leads to dysbiosis (lack of normal intestinal homeostasis), obesity (increased Firmicutes to Bacteroidetes ratio), inflammatory bowel disease (decreased Firmicutes to Bacteroidetes ratio), and other diseases [113]. The Firmicutes phylum includes *Clostridium* (95% of this phylum), *Lactobacillus*, *Bacillus*, *Enterococcus*, and *Ruminicoccus* genera, and the Bacteroidetes phylum consists of *Bacteroides* and *Prevotella* genera [114]. Although early studies estimated the microorganism population as more than 100 trillion and number of human cells as around 10 trillion, more recent estimates state a ratio of 1.3 bacteria cells to each human cell [115]. The microbiome produces a wide variety of metabolites and can account for some of the variation in plasma metabolites between individuals [116]. The composition of the gut microbiome and gut-derived metabolites are associated with the occurrence of a wide variety of chronic diseases [117]. In addition, the effect that diet and exercise have on cognition is affected by the gut microbiome [118]. Furthermore, the microbiota was found to affect social behavior in zebrafish during early neurodevelopment [119]. However, the gut microflora can be affected by various factors including consumption of fermented dairy products [120–122].

While acute (high-grade but short-term) inflammation is needed for healing, trigger removal, and tissue repair, systemic chronic (low-grade but persistent) inflammation can lead to a wide variety of adverse health conditions including metabolic syndrome (hypertension, hyperglycemia, and dyslipidemia), type 2 diabetes, nonalcoholic fatty liver disease, cardiovascular disease, chronic kidney disease, multiple cancer types, depression, neurodegenerative and autoimmune diseases, osteoporosis, and sarcopenia [123]. Probiotics, along with prebiotics, resistant starch, and resistant proteins, can decrease chronic low-grade inflammation by producing short-chain fatty acids (acetate, propionate, and butyrate), improving phagocytic activity, and reducing pro-inflammatory cytokine production to potentially promote healthy aging [124].

Probiotics provide many health benefits. Some of these health benefits provided by probiotics, postbiotics, and paraprobiotics (to be discussed later) with either mixed or strong evidence for effectiveness in clinical trials are summarized in Table 3 [125–225]. Because of the complexity involved in being consistent when evaluating the strength of the evidence for the effectiveness of probiotics in preventing or treating each of these adverse health conditions or providing the health benefits, no attempt was made for this evaluation. The efficacy of probiotics in controlling Crohn's disease usually could not be shown [226]. More details about the health benefits provided by yogurt and probiotic fermented milks are provided by Sakandar and Zhang [227], and Hadjimbei et al. [228].

**Table 3.** Some health benefits for which probiotics, postbiotics, and paraprobiotics have shown a mixed to favorable result in an original study or in a meta-analysis. Due to the difficulty of being consistent involved in evaluating the strength of the evidence for the effectiveness of probiotics in preventing or treating each of these health conditions, no attempt was made for the evaluation of effectiveness for the probiotics listed in this table.

| Health Condition | Probiotic | Original Article or Review Paper | Reference |
|---|---|---|---|
| Periodontal disease | | Review | [125] |
| Bacterial tonsillitis | *Streptococcus salivarius* BIO5 | Original | [54] |

**Table 3.** *Cont.*

| Health Condition | Probiotic | Original Article or Review Paper | Reference |
|---|---|---|---|
| Anti-inflammatory and antibiofilm activities against oral pathogens | *Enterococcus faecalis* M157 in fermented whey | Original | [126] |
| Lactose intolerance | | Review | [127] |
| Galactosemia | Galactose positive *S. thermophilus* NCDC 659 (AJM), 660 (JMI), and 661 (KM3) | Original | [128] |
| Short-chain fatty acid production | VSL#3 [1] | Original | [129] |
| Vitamin production | | Review | [130] |
| Gamma-aminobutyric acid production | *L. plantarum* K16 | Original | [131] |
| Protection against foodborne illness | | Review | [132] |
| Colonization of *Campylobacter* | *L. plantarum* LPS | Original | [133] |
| Anti-listerial activity | Postbiotics of *L. acidophilus* LA5, *L. casei* 431, and *L. salivarius* Ls-BU2 | Original | [134] |
| Antimicrobial therapy | | Review | [135] |
| Gut microbiome development in very preterm infants | Either *B. bifidum* and *L. acidophilus* or *B. bifidum* and *B. longum* subsp. *infantis* and *L. acidophilus* | Original | [136] |
| Healthy microbiome | *B. subtilis* DE111 | Original | [137] |
| Restoration of microbiome after antibiotic treatment | *L. acidophilus* and *B. bifidum* | Original | [138] |
| Improve microbiome in cirrhosis patients | Multispecies probiotics | Original | [139] |
| Modulate gut microbiota and reduce exposure to uremic toxins in hemodialysis patients | Bifico (*B. longum* NQ1501, *L. acidophilus* YIT2004, and *E. faecalis* YIT0072) | Original | [140] |
| Gut bacterial diversity | *Bacillus coagulans* GBI-30 6086 | Original | [141] |
| Leaky gut | Probiotic cocktail of 5 *Lactobacilli* and 5 *Enterococci* strains | Original | [142] |
| Improve Gut Epithelial Barrier | *S. thermophilus* BGKMJ1-36 and *L. bulgaricus* BGVLJ1-21 | Original | [143] |
| Antioxidative activity | | Review | [144] |
| Antioxidant activity and intestinal permeability in cancer carcinogenesis | VSL#3 [1] | Original | [145] |
| Oxidative and inflammatory stress reduction | *L. plantarum* S1 (viable and heat-killed cells and metabolites) from fermented whey | Original | [146] |
| Immunity | | Review | [147] |
| Exopolysaccharide production for immunomodulatory, antimicrobial, antioxidant, and anticancer activities | *Lactobacillus* | Review | [148] |
| Highly symptomatic celiac disease | *Bifidobacterium infantis* NLS super strain | Original | [149] |
| Viral infections | Various probiotics and paraprobiotics | Review | [150] |
| Possible inhibition of HIV transmission and replication | Engineered *L. rhamnosus* GG and GR-1 | Original | [151] |
| Diarrhea in HIV/AIDS patients | Probiotic yogurt with *L. rhamnosus* GR-1 and *L. reuteri* RC-14 | Original | [152] |
| Antibiotic-associated diarrhea | *B. animalis* subsp. *lactis* XLTG11 | Original | [153] |
| Chemotherapy-induced diarrhea in lung cancer patients | *Clostridium butyricum* | Original | [154] |
| Enteral-tube-feeding diarrhea [2] | | Review | [155] |
| Childhood rotavirus infections | | Review | [156] |
| Acute pediatric diarrhea | | Review | [157] |
| Travelers diarrhea | *Lactobacillus* GG | Original | [158] |
| | *L. acidophilus* and B. *bifidum* | Original | [159] |
| *Clostridioides difficile* diarrhea | *L. rhamnosus* GG | Original | [160] |
| *Helicobacter pylori* infection | *Limosilactobacillus fermentum* UCO-979C | Original | [161] |

| Health Condition | Probiotic | Original Article or Review Paper | Reference |
|---|---|---|---|
| Constipation | *L. acidophilus* LA11-Onlly, *L. rhamnosus* LR22, *L. reuteri* LE16, *L. plantarum* LP-Onlly, and *B. animalis* subsp. *lactis* BI516 | Original | [162] |
| | *L. rhamnosus* LR-168, *L. acidophilus* LA-99, and *B. animalis* BB-115 | Original | [163] |
| Irritable bowel syndrome | | Review | [164] |
| Necrotizing enterocolitis | *B. longum* subsp. *infantis* | Original | [165] |
| Ulcerative colitis | | Review | [166] |
| | | Review | [167] |
| Hospital stay for acute pancreatitis | | Review | [168] |
| Colorectal cancer | | Review | [169] |
| Gastrointestinal cancer | | Review | [170] |
| Liver and breast cancer | *Streptococcus salivarius* BP8, BP156, and BP160 | Original | [171] |
| Breast cancer | | Review | [172,173] |
| Prostate cancer | Whey beverages with *L. acidophilus* La-05, *L. acidophilus* La-03, *L. casei-01*, and *B. animalis* Bb-12 | Original | [174] |
| Cervical cancer | | Review | [175] |
| Polycystic ovary syndrome | | Review | [176] |
| Vaginosis | *Lactobacillus* | Original | [177] |
| Antimicrobial activity (hydrogen peroxide, bacteriocins, and lactic acid production) for vaginal health | *Lactobacillus crispatus* | Review | [178] |
| Inhibit sperm activity | *Lactobacillus crispatus* | Original | [179] |
| Male fertility disorders | | Review | [180] |
| Bladder cancer | | Review | [181] |
| Bladder diseases (bladder cancer, interstitial cystitis, and overactive bladder) | | Review | [182] |
| Reduce exposure to uremic toxins in hemodialysis patients | Bifico (*B. longum* NQ1501, *L. acidophilus* YIT2004, and *E. faecalis* YIT0072) | Original | [140] |
| Pediatric urinary tract infection recurrence | *L. acidophilus*, *L. rhamnosus*, *B. bifidum*, and *B. lactis* | Original | [183] |
| Urinary excretion of oxalate (risk factor for renal stones) | *L. acidophilus*, *L. brevis*, *L. plantarum*, *B. infantis*, and *S. thermophilus* | Original | [184] |
| Idiopathic nephrotic syndrome | *Clostridium butyricum* | Original | [185] |
| Lung metastasis of melanoma cells | VSL#3 [1] | Original | [129] |
| Respiratory tract infection | | Review | [186] |
| Influenza A virus | *L. mucosae* 1025 and *B. breve* CCFM1026 | Original | [187] |
| COVID-19 | Probiotics and their metabolites | Review | [188] |
| Ventilator-associated pneumonia in critically ill patients | | Review | [189] |
| Allergic rhinitis | *Bifidobacterium* mixture | Review | [190] |
| Respiratory allergy | Commercial probiotic fermented milk | Original | [191] |
| Asthma | *L. paracasei* K47 | Original | [192] |
| Cystic fibrosis | | Review | [193] |
| Atopic dermatitis | | Review | [194] |
| Skin disorders (atopic dermatitis, psoriasis, rosacea, and acne vulgaris) | | Review | [195] |
| Skin health | *L. reuteri* ATCC 6475 | Original | [196] |
| Dry eye | *L. plantarum* NK151 and *B. bifidum* NK175 | Original | [197] |
| Vernal keratoconjunctivitis | *L. acidophilus* eye drops | Original | [198] |
| Rheumatoid arthritis [2] | | Review | [199,200] |
| Recovery from bone fractures | *L. casei* Shirota | Original | [201] |
| Pain relief after rib fracture | *L. casei* Shirota | Original | [202] |
| Mineral absorption and bone health | *L. rhamnosus* HN001 | Original | [203] |

**Table 3.** *Cont.*

| Health Condition | Probiotic | Original Article or Review Paper | Reference |
| --- | --- | --- | --- |
| Calcium absorption | *L. rhamnosus* GG * | Original | [204] |
| Iron absorption | | Review | [205] |
| Blood lipids | *B. subtilis* DE111 | | [206] |
| Fasting glucose and insulin levels | | Review | [207] |
| Diabetes (blood pressure, fasting blood sugar, cholesterol, triglyceride, hemoglobin A1c, high sensitive C-reactive protein) | Probiotic yogurt | Original | [208] |
| Serum triglyceride and glucose | *Bacillus coagulans* GBI-30 6086 | Original | [141] |
| Atherosclerosis (lesion formation, dyslipidemia, endothelial dysfunction, inflammation, hypertension and hyperglycemia, and TMAO (trimethylamine *N*-oxide)) | | Review | [209] |
| Infantile colic | *B. breve* CECT7263 | Original | [210] |
| Obesity | *L. reuteri* ATCC 6475 | Original | [211] |
| | | Review | [212] |
| Liver fibrosis | *L. paracasei*, *L. casei*, and *Weissella confusa* | Original | [213] |
| Non-alcoholic fatty liver disease | | Review | [214] |
| Hyperuricemia | | Review | [215] |
| Phenylketonuria | Genetically engineered probiotics | Review | [216] |
| Exercise performance and decrease fatigue | *L. salivarius* subsp. *salicinius* SA-03 | Original | [217] |
| Sleep | | Review | [218] |
| Depression and anxiety | | Review | [219] |
| Anxiety | | Original | [220] |
| Serotonin biosynthesis from tryptophan | *L. plantarum* LRCC5314 | Original | [221] |
| Mood | | Original | [222] |
| Memory and learning | *L. paracasei* ssp. *paracasei* BCRC 12188, *L. plantarum* BCRC 12251, and *S. thermophilus* BCRC 13869 | Original | [223] |
| Age related dementia | | Review | [224] |
| Autism | | Review | [225] |

[1] VSL#3 includes *B. breve*, *B. infantis*, *B. longum*, *L. acidophilus*, *L. bulgaricus*, *L. casei*, *L. plantarum*, and *S. thermophilus*.
[2] Mixed results. * Inulin was also included in the treatment which may have contributed to the favorable results.

Different strains of bacteria provide their health benefits by different mechanisms [229], and knowledge of these mechanisms can help in probiotic selection and modification for effectively treating disease. Four main mechanisms by which probiotics confer health benefits include potential pathogen interference, barrier function improvement, immunomodulation, and neurotransmitter production [230]. Pathogen interference mechanisms include production of antimicrobial compounds including bacteriocins and defensins, competition with pathogens, inhibition of adherence of pathogens, and luminal pH reduction [229]. Probiotics such as *L. rhamnosus* can be bioengineered for an alternative method for pathogen inhibition within the field known as pathobiotechnology [231].

Gut microbiomes vary from person to person [114]. Individuals vary in the ability of which consumed probiotics, such as in a fermented milk product, are able to modify the composition of the autochthonous gut microflora, suggesting that a tailored diet may be needed for individuals that are on a beneficial microbial based therapy and have a resistant gut microbiota [232]. Veiga et al. [233] predicts that many people will have their genome sequenced in the future that will allow them to tailor specific probiotics (referred to as precision probiotics) to their unique human-microbiome symbiosis to optimize their microbiome-centered nutrition and preventative health care. Perhaps in the future, yogurt could be a carrier for these precision probiotics.

## 5. Probiotic Strains Used in Yogurt and Related Probiotic Milk Beverages and Their Health Benefits

One question a product developer working with yogurt needs to ask is whether to use a single strain probiotic or to use multiple strains for forming potential symbiotic relationships (similar to the relationship between *S. thermophilus* and *L. bulgaricus*) or for potential health benefits. Peng et al. [234] manufactured yogurt with incorporated *L. casei* Zhang, *B. lactis* V9, or their combination. The use of this combination stimulated the growth of *B. lactis* V9 compared to use of *B. lactis* V9 by itself, likely because of valine, leucine, and isoleucine biosynthesis. However, the use of this combination did not stimulate the growth of *L. casei* Zhang compared to the use of *L. casei* Zhang by itself. Furthermore, the use of this combination stimulated short-chain fatty acid production. In a similar type of experiment, Fan et al. [50] manufactured yogurt with incorporated *Lacticaseibacillus casei* CGMCC1.5956, *Levilactobacillus brevis* CGMCC1.5954, or their combination. They found improved probiotic growth, increased hardness and adhesiveness, less syneresis, and enhanced antioxidant capacity in the yogurt prepared with both probiotics. In another study, Fan et al. [235] found that use of binary probiotics (*Lacticaseibacillus casei* CGMCC1.5956 and *Lactiplantibacillus plantarum* subsp. *plantarum* CGMCC1.5953) enhanced hardness, viscosity, and gumminess compared to use of these probiotics by themselves. McFarland [236] reviewed whether single strains or multiple strains are more effective in preventing and treating diseases. Although there were cases in which multiple strains were more effective than single strains in eradicating diseases, multi-strain mixtures were not usually more effective than single strain probiotics. However, Washburn et al. [237] reported that microbial gastrointestinal diversity was not significantly influenced in their study when healthy adults consumed *Bifidobacterium infantis* as a single probiotic species.

It has been debated as to whether or not the yogurt starter cultures, *S. thermophilus* and *L. bulgaricus*, should be considered as probiotic [238]. Obviously, if the yogurt is heat treated to kill the cultures after fermentation, then it is not probiotic yogurt. One would not consider yogurt starter cultures as probiotic if they were not acid and bile tolerant or if they did not survive within the intestinal tract [238]. In an early study, Cheplin and Rettger [239] were not able to implant *Bacillus bulgaricus* (now *L. bulgaricus*) into the human gastrointestinal tract. However, Mater et al. [240] and Elli et al. [241] found that *L. bulgaricus* and *S. thermophilus* can survive within the human gastrointestinal tract. Martinović et al. [242] reviewed whether or not *S. thermophilus* survives within the gastrointestinal tract and concluded that most studies did not perform taxonomic studies with sufficient accuracy for distinguishing *S. thermophilus* from *S. salivarius* to determine if *S. thermophilus* can be recovered. Uriot et al. [243] supported the idea that certain strains of *S. thermophilus* be considered as probiotic, and Guarner et al. [244] likewise concluded that both *S. thermophilus* and *L. bulgaricus* should be considered as probiotic. Popović et al. [143] showed that *S. thermophilus* BGKMJ1-36 and *L. bulgaricus* BGVLJI-21 can function as yogurt starter cultures and possess probiotic properties by modulating gut autophagy and improving the gut epithelial barrier. Recently, Taj et al. [245] found that certain exopolysaccharide producing strains (RIRT2, RIH4, and RIY) of *S. thermophilus* fulfill the basic criteria to be considered as probiotics. In this paper, the emphasis is on yogurt that has probiotic cultures in addition to *S. thermophilus* and *L. bulgaricus*.

*Lactobacillus acidophilus* NCFM (North Carolina Food Microbiology) (ATCC 700396) has been available since 1972 and is GRAS (GRN Number 357). The complete genomic sequence has been published by Altermann et al. [246]. This strain is of human origin and was isolated and characterized by Gilliland et al. [247]. This probiotic was reviewed by Sanders and Klaenhammer [248]. Health benefits provided by *L. acidophilus* NCFM include antimicrobial activity against foodborne pathogens, in vitro evidence for adherence to human cells, possible ability to assimilate cholesterol from lab growth media, ability to survive within the gastrointestinal tract and to be isolated in human feces, possess active lactase to possibly assist with lactose digestion, beneficially effect colonization in the small bowel, decreasing potentially harmful microbial activities related to cancer development

in the intestine, improve immune response when combined with other yogurt cultures to oral antigens, potentially controlling urogenital infections in women, potentially reducing incidence of diarrhea when combined with other probiotics, and protecting against systemic infections [248].

*Lactobacillus acidophilus* LA-5 provides many health benefits and has been incorporated into food and dietary supplements since 1979 [249]. This strain is frequently investigated with *Bifidobacterium animalis* subsp. *lactis* BB-12 which will be discussed below. When combining *L. acidophilus* LA-5 with *B. animalis* subsp. *lactis* BB-12, improved suppression of *Helicobacter pylori* infections [250], improved relief from chronic constipation [251], reduced symptoms and inflammation from ulcerative colitis [252], better recovery to colonic surgery as part of an optimization package [253], improved glycemic control and antioxidant status in patients with Type 2 diabetes [254,255], and reduced oxidative stress [256] have been reported. *L. acidophilus* LA-5, *B. animalis* subsp. *lactis* BB-12, and *L. casei* TMC incorporated into milk reduced total cholesterol and low-density lipoprotein cholesterol levels in mild hypercholesterolemic study volunteers [257]. *L. acidophilus* LA-5 produces lactacin B bacteriocin (peptides that inhibit certain other types of bacteria within the same environment) in the presence of yogurt starter cultures [258]. Furthermore, *L. acidophilus* LA-5 can produce conjugated linoleic acid in supplemented cheese whey [259] and attenuate obesity [260].

*Lactobacillus helveticus* has been used as a probiotic in yogurt. Zhou et al. [39] manufactured yogurt using *L. helveticus* H9 as an adjunct starter culture and found shortened fermentation time, increased richness of volatile flavor compounds, production of fermented milk antihypertensive peptides Val-Pro-Pro and Ile-Pro-Pro, but lower sensory scores compared to their control yogurt. Kajimoto et al. [261] manufactured a liquid yogurt that contains lactotripeptides and the starters *L. helveticus* and *Saccharomyces cerevisiae,* and found that consumption of this product decreased systolic and diastolic blood pressure significantly more than the placebo group in mild hypertensive subjects in a placebo-controlled, double-blind study. Yamamura et al. [262] reported that milk fermented with *L. helveticus* strain CM4 may improve sleep in healthy elderly Japanese subjects.

*Lacticaseibacillus casei* (formerly *Lactobacillus casei*) DN-114001 (*L. casei* strain CNCM I-1518 or *L. casei* Immunitas®) is used in the probiotic yogurt-like drink DanActive (Actimel) and has been patented [263]. *L. casei* DN-114001 was effective in increasing fecal *Bifidobacterium* counts but decreasing fecal *Clostridium* counts in children [264] and may be effective in reducing atopic dermatitis in children [264,265]. Agarwal et al. [266] reported that *L. casei* DN-114001 was effective in controlling diarrhea in people from developing countries and using *L. casei* DN-114001 as a starter for producing dahi was also effective. Guillemard et al. [267] found a reduced average and cumulative duration of common infectious diseases and reduced episodes and cumulative durations of upper respiratory tract infections and rhinopharyngitis in the free-living elderly upon consumption of a fermented dairy product containing *L. casei* DN-114001. Marcos et al. [268] studied anxiety levels and immune responses of academically-stressed university students who consumed fermented milk containing *L. casei* DN-114001 versus a control. Although there was no significant difference in treatment effect in anxiety, they found that consumption of this fermented drink modified the number of lymphocytes and CD56 cells in the stressed students [268].

*Lacticaseibacillus paracasei* (formerly *Lactobacillus casei*) Shirota strain is used in a fermented milk drink called Yakult® and is GRAS (GRN Number 429). This product was launched in 1935 and is fermented by this probiotic until a titratable acidity of 2% is reached [269]. The resulting curd is broken, sweetened and flavored, homogenized at 15 MPa, and diluted with water before being packaged into 65-mL plastic bottles. This strain is indigenous in the human intestinal tract [270]. Yasuda et al. [271] analyzed genes in *L. casei* strain Shirota (YIT 9029) related to synthesis of polysaccharides associated with the cell wall involved in regulating host immunity based on their unpublished in-house data of the complete genome sequence of this probiotic. Kato-Kataoka et al. [272] reported

that medical students who daily consumed fermented milk containing *L. casei* strain Shirota displayed fewer physical symptoms when exposed to stressful academic examinations.

*Lacticaseibacillus rhamnosus* GG (Gorbach and Goldin) (formerly *Lactobacillus rhamnosus* GG and *Lactobacillus acidophilus* GG and also called LGG) (ATCC accession number 53103) (GRN Numbers 845 and 1013) is a widely studied probiotic. It has been patented [273,274] and its complete genome sequence has been published [275]. Capurso [276] reviewed the effect of *L. rhamnosus* GG on gastrointestinal infections and diarrhea, antibiotic and *Clostridium difficile* associated diarrhea, irritable bowel syndrome, inflammatory bowel disease, respiratory tract infections, allergy, cardiovascular diseases, nonalcoholic fatty liver disease, nonalcoholic steatohepatitis, cystic fibrosis, cancer, exercise physiology, and the elderly. Szajewska and Hojsak [277] concluded that the symptoms of acute gastroenteritis can be managed and antibiotic-associated diarrhea can be prevented upon administering *L. rhamnosus* GG to toddlers and older children.

*Limosilactobacillus* (formerly *Lactobacillus*) *reuteri* DSM 17,938 is GRAS (GRN Number 254), and its use in treating various pediatric gastrointestinal disorders has been recently reviewed [278]. This species inhibits pathogen growth by secreting reuterin and other substances and restores homeostasis by interacting with intestinal microbiota and mucosa. Furthermore, this species can increase the pain threshold and gastrointestinal motility. The duration of acute diarrhea and hospitalization for acute gastroenteritis can be decreased. Likewise, antibiotic-associated side effects from treating *Helicobacter pylori* infections can also be decreased with this species [278].

*Lactiplantibacillus plantarum* (formerly *L.actobacillus plantarum*) can successfully be used in producing probiotic yogurt. Li et al. [279] added various strains of *L. plantarum* to produce yogurt. They found that milk as a medium is appropriate as a carrier for *L. plantarum* because of their survival both during fermentation and storage. No negative sensory quality effects were found. Strain IMAU 70,095 of *L. plantarum* was found to be the most suitable strain for yogurt [279]. *L. plantarum* CCFM47 and CCFM232, *L. acidophilus* CCFM6, and *L. rhamnosus* GG were able to inhibit α-glucosidase and survive at rates up to 60% in simulated gastrointestinal juices [280]. Furthermore, viability of *L. plantarum* CCFM47 and *L. acidophilus* CCFM6 in yogurt was improved when yogurt was supplemented with soybean oligosaccharides, indicating that this type of yogurt would be expected to have antihyperglycemic properties.

*Bifidobacterium animalis* subsp. *lactis* BB-12 is a probiotic that is GRAS (GRN Number 856). It is commonly studied with *L. acidophilus* LA-5. Garrigues et al. [281] published the complete genome sequence of *B. animalis* subsp. *lactis* BB-12. This microorganism can survive within the gastrointestinal tract with excellent resistance to acid and bile and supports a healthy gastrointestinal microbiota. Also, it improves bowel function, protects against diarrhea in infants and children, and reduces antibiotic-associated diarrhea. Additionally, this microorganism provides protection against respiratory tract infections [282]. A potential anti-inflammatory effect was found in healthy adults upon consumption of yogurt with incorporated *B. animalis* subsp. *lactis* BB-12 [283].

*Bifidobacterium animalis* subsp. *lactis* HN019 has been patented (US patent 6379663) [284] and is GRAS (GRN Number 445). Its complete genome sequence has been described in Morovic et al. [285]. Magro et al. [286] found that the colonic transit time for constipated patients consuming yogurt containing *B. lactis* HN019, *L. acidophilus* NCFM, and polydextrose was shortened compared to control yogurt. Likewise, Miller et al. [287] analyzed 15 clinical trials representing 675 subjects for determining the effectiveness of various probiotics in reducing intestinal transit times and found that *B. animalis* subsp. *lactis* HN019 and *Bifidobacterium animalis* DN-173 010 were the most effective probiotic strains. Gut health benefits provided by *B. animalis* subsp. *lactis* HN019 have been reviewed by Cheng et al. [288].

*Bifidobacterium animalis* subsp. *lactis* DN-173 010/CNCM I-2494 is a probiotic that is added to various forms of the Activia brand of yogurt and drinks (Groupe Danone) that was launched in France in 1987 for relieving minor digestive discomfort. Studies have

shown that consumption of fermented milk containing *B. animalis* DN-173 010 increased stool frequency for constipated children [289], improved health-related quality of life with increased stool frequency (in a subgroup of subjects with fewer than three bowel movements per week) and with decreased bloating in constipated adult subjects with irritable bowel syndrome [290], and decreased gut transit times in elderly people [291]. Reducing gastrointestinal inflammation in inflammatory bowel disease, ulcerative colitis, and/or Crohn's Disease or preventing or treating irritable bowel syndrome by having a subject consume *B. animalis* or Activia® have been patented (US patent 8,685,388 B2) [292].

*Weizmannia coagulans* (formerly *Bacillus coagulans*) and *Bacillus subtilis* are probiotics that can be used in yogurt. Ma et al. [57] found that *W. (B.) coagulans*-70 was a desirable strain when used as an adjunct starter culture due to its high count in yogurt during fermentation and storage, the increased in yogurt hardness and viscosity during storage, and high sensory evaluation scores of the yogurt samples. *W. (B.) coagulans* GBI-30, 6086 (BC30™) has been patented [293] and has FDA GRAS status (GRN Numbers 399 and 670). *W. (B.) coagulans* was originally called *Lactobacillus sporogenes*, but this nomenclature is not correct [294] since lactobacilli do not form spores. *Bacillus subtilis* can be successfully added to yogurt, and peptides that are present in this yogurt have a high antioxidant potential and can improve shelf-life [295].

## 6. Prebiotics, Synbiotics, Paraprobiotics, Postbiotics, and Psychobiotics

The definition of prebiotic has evolved over time. Gibson and Roberfroid [296] defined prebiotic as "non-digestible food ingredients that beneficially affects the host by selectively stimulating the growth and/or activity of one or a limited number of bacterial species already resident in the colon". More recently, the definition of prebiotic has been modified to "a substrate that is selectively utilized by host microorganisms conferring a health benefit" [297]. Examples of prebiotics include inulin, fructo-oligosaccharides, galacto-oligosaccharides, isomalto-oligosaccharides, human milk oligosaccharides, xylo-oligosaccharides, xylan, lactulose, oat fiber (β-glucan), pectin, guar gum, resistant starch, stachyose, select polyphenols, bacterio-phage, omega-3 fatty acids, and yeast hydrolysate [219,298,299]. Prebiotics are discussed in detail by Gibson and Roberfroid [300].

Synbiotics were defined by a panel from the International Scientific Association for Probiotics and Prebiotics as "a mixture comprising live microorganisms and substrate(s) selectively utilized by host microorganisms that confers a health benefit on the host" [301]. Therefore, the proper type of prebiotic must be used with a given probiotic. Stronger health benefits occur when a product contains synbiotics rather than either a probiotic or prebiotic alone [302]. Dairy products incorporating synbiotics in research studies include yogurt and a yogurt-based drink, fermented skim milk, cheeses, ice cream, and infant formula. Some of the probiotics and prebiotics that have been incorporated into synbiotic yogurt are listed in Table 4. For non-dairy based products, this list includes bread buns, chocolate, candy, mousse, Andean blackberry slices, soybean beverage, fermented soy food, cereal mix, traditional Indian dry snack, dry malted drink, and salad dressing [302].

**Table 4.** Some of the probiotics and prebiotics that have been incorporated into synbiotic yogurt.

| Probiotic | Prebiotic | Reference |
|---|---|---|
| *L. acidophilus* ATCC 4357 | Fructooligosaccharide and Isomaltooligosaccharide | [303] |
| *L. acidophilus* LA-5 | Oligofructose-enriched inulin | [304] |
| *L. acidophilus* LA-5 | Honey and aqueous cinnamon extract | [305] |
| *L. acidophilus* ATCC 4356 | Flaxseed | [306] |
| *L. acidophilus* | White oyster mushroom flour and Taro flour | [307] |
| *L. acidophilus* 100021 and *L. helveticus* 501699 | Inulin and maltodextrin | [308] |
| *L. acidophilus* LA-5 and *B. animalis* subsp. *lactis* BB-12 | Monk fruit extract, inulin, and pectin | [309] |
| *L. acidophilus* ATCC 4356 and *B. longum* | Purple Sweet Potato | [310] |
| *L. acidophilus* and *B. animalis* subsp. *lactis* | Black carrot pulp and Exudate acacia gum | [311] |

**Table 4.** *Cont.*

| Probiotic | Prebiotic | Reference |
|---|---|---|
| *L. acidophilus* and *Bifidobacteria* | Citrus peels of sour orange, sweet orange, and lemon | [312] |
| *L. acidophilus* and *L. casei* | Stachyose | [313] |
| *L. casei* 01 | Inulins of varying chain lengths: short (P95), medium (GR) and long (HP) | [314] |
| *L. casei* 01 | Inulin, polydextrose, and modified starch | [315] |
| *L. casei* strain Shirota | Inulin or fructans from *Agave salmiana* Otto ex Salm-Dyck | [316] |
| *L. casei* and *L. gasseri* | Banana fiber and peel banana fiber | [317] |
| *L. casei* 431, *L. rhamnosus* LGG, and *B. animalis* subsp. *lactis* BB-12 | Banana peel powder and Mango peel powder | [318] |
| *L. rhamnosus* LGG | Inulin | [204] |
| *L. paracasei* [1] | Lactitol [1] | [319] |
| *L. paracasei* | Galactofructose, inulin, soy protein isolate, and spirulina | [320] |
| *L. plantarum* CFR 2194 and *L. fermentum* CFR 2192 | Fructooligosaccharides | [321] |
| *L. plantarum* [1] | Sorbitol | [322] |
| *L. plantarum* TISTR 1465 | Black waxy rice | [323] |
| *L. reuteri* DPC16 | Manuka honey | [324] |
| *B. longum* ATCC 15707 | Rosemary extract | [325] |
| *B. animalis* Bb-12 | Inulin | [56] |
| *B. infantis* Bb02 | Gluten Friendly Flour™ | [326] |
| *Propionibacterium freudenreichii* ssp. *shermanii* (PS-4) | Inulin | [327] |

[1] Synbiotic microcapsules.

Paraprobiotics (ghost probiotics) are "non-viable microbial cells (intact or broken) or crude cell extracts (i.e., with complex chemical composition), which, when administered (orally or topically) in adequate amounts, confer a benefit on the human or animal consumer" [328]. A wide variety of probiotics can be used for paraprobiotic production [328,329]. Techniques for inactivating probiotics to produce paraprobiotics include the use of irradiation, high pressures, sonication and high-intensity ultrasound, ultraviolet rays, ohmic heating, pulsed electric field, supercritical carbon dioxide, drying, pH changes, and thermal treatments [329,330]. Paraprobiotics can be produced by ohmic heating at electric field magnitudes of 8 V/cm for inactivating probiotic cultures (*Lactobacillus acidophilus* LA-5, *Lacticaseibacillus casei* 01 and *Bifidobacterium animalis* subsp. *lactis* Bb-12) [331]. However, ohmic heating performed at sub-lethal conditions can increase cellular permeability and improve nutrient absorption leading to faster cellular growth [332]. High-intensity ultrasound can also be used to inactivate probiotics for generating paraprobiotics by adjusting process parameters resulting in a high level of sonoporation within the cell, leakage of cellular content, and fluctuation of the cell membrane lipid bilayer [330]. Health benefits provided by the use of paraprobiotics include immunomodulation, pathogen inhibition, intestinal microbiota modulation, intestinal injury recovery, reduction of bacterial translocation and preservation of the intestinal barrier, treatment of diarrhea, inflammation modulation, reduction of lactose intolerance, cholesterol reduction, respiratory disease reduction, treatment of alcohol-induced liver diseases, cancer growth inhibition, atopic dermatitis treatment, visceral pain response modulation, colitis treatment, suppression of certain age related diseases, and dental caries inhibition [329].

Paraprobiotics have been incorporated into yogurt. Parvarei et al. [333–335] incorporated heat-killed or viable *L. acidophilus* ATCC SD 5221 or heat-killed or viable *B. lactis* BB-12 into yogurt either before or after fermentation and compared the properties to a control yogurt. Viability of starter cultures was increased with the addition of paraprobiotics [333]. They typically found less syneresis and greater water holding capacity for yogurts containing the paraprobiotics added before fermentation compared to the remaining yogurts due to the presence of exopolysaccharides derived from the inactivated cells [333]. There were pores and void spaces within the microstructure of these yogurts containing paraprobiotics

added before fermentation [334]. Yogurts incorporating heat-killed *B. lactis* BB-12 into yogurt before fermentation had the highest flavor and mouthfeel sensory scores [335].

Postbiotics have been defined by a panel from the International Scientific Association for Probiotics and Prebiotics as a "preparation of inanimate microorganisms and/or their components that confers a health benefit on the host" [336]. Guimarães et al. [330] suggested that postbiotics can be produced by high-intensity ultrasound. Advantages of using postbiotics are their inability to cause infections because they are not alive and their long shelf-life. Depommier et al. [337] administered pasteurized (killed) *Akkermansia muciniphila* to individuals who were overweight or obese with insulin resistance and found improved insulin sensitivity but reduced insulinemia and plasma total cholesterol. Darwish et al. [338] produced a functional yogurt incorporating the postbiotic *E. coli* Nissle 1917 and Cape gooseberry, and increased the antimicrobial, antitumor, and antioxidant activities and total phenolic content of the yogurt.

Zendeboodi et al. [339] proposed a new classification of probiotics as true probiotics meaning viable and active, pseudoprobiotics meaning viable and inactive either in the form of vegetative cell or spore, and ghost probiotics meaning nonviable cell either in the form of intact or ruptured cells.

Psychobiotics have been defined as "a live organism that, when ingested in adequate amounts, produces a health benefit in patients suffering from psychiatric illness" [340]. There is bilateral communication between the gut microbes and the brain, and this association is known as the gut–brain–microbiota axis. Although gut dysbiosis (an imbalance of the microorganisms in the gastrointestinal tract as discussed earlier) can lead to altered brain function, mental illness (including major depressive disorder and schizophrenia), and neurological decline (such as Alzheimer's disease) in old age [341], certain probiotics may provide positive mental health effects. These probiotics can produce neuroactive substances (including gamma-aminobutyric acid and serotonin) that affect the brain–gut axis and provide an antidepressant effect [340]. There are many fermented food and beverage applications for these potential psychobiotic strains as summarized by Toro-Barbosa et al. [342]. Benton et al. [222] supplied subjects with Yakult (a milk drink containing the probiotic *L. casei* (now *L. paracasei*) Shirota) and found improvements in the mood of subjects with initially poor moods after they consumed this product. In another study in which subjects consumed *L. casei* (now *L. paracasei*) Shirota, Rao et al. [220] found reduced anxiety among subjects with chronic fatigue syndrome consuming the probiotic compared to the placebo. Furthermore, these researchers found that the probiotic-consuming subjects had a greater increase in *Lactobacillus* and *Bifidobacteria* in their feces compared to the placebo-controlled subjects.

## 7. Probiotic Yogurt Products Currently Available in the Market

Many brands of probiotic yogurt with varying claims can be found on the market. Use of only A2 milk for making yogurt has been claimed. Some brands claim to be made from milk from grass-fed cows that optimize the ratio of omega-6 fatty acids to omega-3 fatty acids and have enhanced conjugated linoleic acids. Other brands claim that the milk used to produce the yogurt is organic or comes from cows not treated with rBST. Fat contents of yogurt of up to 10% have been reported. A reduction of 90% lactose compared to regular yogurt has also been claimed. Honey or chia seed can be found in certain types of commercial yogurt. High protein yogurt is available on the market. Low and slow heating has been declared for one type of yogurt. Some manufacturers do not homogenize their milk used to make their yogurt. A wide variety of probiotics are used in commercial yogurt. *Bacillus coagulans*, which becomes active upon reconstitution with water, is used in a brand of frozen yogurt. One yogurt culture manufacturer claims that over 300 probiotic strains are present in their product. A shelf-life of 11 weeks has been claimed for a yogurt made with *L. bulgaricus* G-LB-44, a powerful pathogen inhibitor. Types of commercial probiotic yogurt include conventional spoonable yogurt, Greek yogurt, Bulgarian yogurt, Balkan yogurt, and frozen yogurt. Yogurts containing postbiotics are also on the market.

## 8. Various Forms of Yogurt

There are many different types of yogurt in addition to spoonable yogurt. Baker [343] has patented a procedure for producing a low calorie, low-fat fruit and *L. acidophilus* - containing yogurt that has the appearance, texture, and taste similar to a conventional fruit-containing yogurt. Pachekrepapol et al. [344] used β-galactosidase enzyme to hydrolyze the lactose found in milk to produce a lactose free, probiotic yogurt with incorporated fructooligosaccharides. Drinkable yogurt may contain a wide variety of probiotic strains including *L. reuteri* WHH1689 [345], *L. gasseri* [346], *L. rhamnosus* HN001 [347], or *L. plantarum* DSM 20205 and *P. acidilactici* DSM 20238 [348] or a combination of probiotics (*Bifidobacterium lactis* Bb-12 and *Lactobacillus acidophilus* LA-5) and prebiotics (soluble corn fiber, polydextrose, and chicory inulin) [349]. A shelf-stable yogurt can be produced by UHT sterilizing milk with a lactose content reduced to about 1% to 1.5%, cooling this product, adding lactic acid bacteria including probiotics, aseptically packaging the product into a container, and storing it under suitable conditions to allow $1 \times 10^7$ cfu/g to $3 \times 10^9$ cfu/g live bacteria to form in the yogurt, resulting in a pH lower than 4.7 [350]. A method for producing high-protein Greek yogurt by concentrating the skim milk by ultrafiltration, combining this concentrated skim milk with other milk fractions to form the yogurt base, fermenting the resulting yogurt base with a yogurt culture and probiotics, and concentrating the fermented product with a ceramic membrane system has been patented [351]. Imer [352] described a process in a patent for producing frozen yogurt by freezing a fermented yogurt mix that may include *L. acidophilus*, *L. casei*, *L. rhamnosus*, and *Bifidobacterium* and incorporating air to an ideal overrun between 30% and 60%. A method of producing Greek frozen yogurt from fermented lactose-reduced skim milk and made without straining, but possibly containing various probiotics, has been described by Bunce and Dave [353]. Natural yogurt that is produced by fermentation by either *L. bulgaricus* or *L. acidophilus* may be dried by various techniques to produce an instant, dry powdered yogurt composition [354]. This composition has a long shelf-life without refrigeration and it can be reconstituted into a yogurt meal or drink. Another patent describes freeze-dried, aerated yogurt that could incorporate prebiotics and probiotics and is readily dissolvable to reduce choking hazard risks [355].

## 9. Use of Probiotic Yogurt as an Ingredient

Probiotic yogurt can be used as an ingredient in the production of other products. Bite sized refrigerated yogurt that can be eaten using fingers can be prepared by coating frozen yogurt portions (possibly containing probiotics) with two layers of fat-based coating [356]. The second layer of this fat coating is applied before allowing the frozen yogurt to thaw, and this second layer may contain particulate inclusions [356]. A snack bar coated with a yogurt containing probiotics (*L. acidophilus* or *B. lactis* or both) and incorporating waxy grains held together by an inulin binder has been patented [357]. A shelf-stable fruit snack that contains an outer layer that could consist of yogurt containing probiotic cultures has been patented [358]. Gutknecht and Ovitt [359] patented low-fat yogurt cheese consisting of 15% to 75% cream cheese, 10% to 40% yogurt incorporating *L. acidophilus*, *Bifidobacterium*, or *L. paracasei* subsp. *casei* in addition to yogurt starter cultures, and 15% to 45% milk protein. Freeze dried yogurt that may contain probiotic cultures is an ingredient in a dry mix food product that also contains other food ingredients (whole grain, fruits, nuts, granola, etc.), and this dry mix can be hydrated to form a thick texture similar to yogurt within 3 min [360]. A shelf-stable light and crunchy yogurt crisp, a snack food, is made from a viscoelastic dough that contains dehydrated yogurt and may contain probiotics, either in the spore form or microencapsulated form [361].

## 10. Useful Functional Ingredients in Probiotic Yogurt

Many ingredients have successfully been added to probiotic yogurt. Some of these useful functional ingredients are listed in Table 5. These functional ingredients include

grains, seeds, flours, fibers, fruits, vegetables, a berry, a nut, juices, spices, essential oils, bee products, and a cyanobacterium.

**Table 5.** Some of the useful functional ingredients that have been incorporated into probiotic yogurt including their concentration and effect on the properties of the resulting yogurt.

| Functional Ingredient Category (in Bold) and Ingredient | Concentration | Effect on Properties | Ref. |
|---|---|---|---|
| **Grain, seed, and flour** | | | |
| Aqueous fennel extract | 2, 4, and 6% | Reconstituting whole milk powder into aqueous fennel extract to manufacture probiotic yogurt resulted in a product with increased phenolic content and antioxidant activity compared to fresh yogurt. | [362] |
| Flaxseed | 0–4% | Flaxseed was successfully added to yogurt containing *L. acidophilus* ATCC 4356. This yogurt had increased *L. acidophilus* counts, viscosity, hardness, cohesiveness, gumminess, and water holding capacity but decreased syneresis and adhesiveness compared to their control yogurt. | [306] |
| Sesame seeds | 6% | Incorporation of roasted sesame into stirred yogurt improveds probiotic viability, sensory properties, and antioxidant properties. | [363] |
| Psyllium husk (Native and acid-modified psyllium husk) | 0.5 g per liter of buffalo milk | Incorporation of psyllium husk into frozen yogurt containing the encapsulated probiotics *L. acidophilus* and *L. plantarum* formed a product with high consumer acceptability. | [364] |
| Oat β-glucan | 0.15% | β-glucan and EPS-producing *B. bifidum* increased viscosity and water holding capacity but decreased syneresis. | [365] |
| Wheat bran | 4% | Incorporation of wheat bran significantly increased total bacterial counts and titratable acidity. | [366] |
| Resisant starch (RS2 and RS3) [1] | 1.5% | This yogurt was made from reconstituted skim milk. RS2 increased serum held within gel network. RS3 protected *B. animalis* subsp. *lactis* BB-12, increased viscosity, and decreased titratable acidity. | [367] |
| Chickpea flour | 0, 1, 2.5, and 5% | Fortification of chickpea flour into probiotic yogurt resulted in improved water holding capacity and decreased syneresis for the resulting yogurt. | [368] |
| **Fiber Ingredient** | | | |
| Inulin of varying chain lengths [2] | 1.5% | P95 lowered the pH but maintained similar flavor scores compared to the control. HP decreased syneresis and improved body and texture compared to the control. | [369] |
| Orange fiber | 0.5, 1, 1.5, and 2% | Incorporating orange fiber into yogurt containing *L. acidophilus* LA-5 and *Bifidobacterium animalis* subsp. *lactis* BB-12 improved antioxidant activity and angiotensin converting enzyme (ACE)–inhibitory activity. | [370] |
| Lemon and orange fibers | 3 g to 200 mL | The enriched fermented milk had good sensory acceptability. *L. acidophilus* and *L. casei* had better survival than *B. bifidum*. | [371] |

**Table 5.** *Cont.*

| Functional Ingredient Category (in Bold) and Ingredient | Concentration | Effect on Properties | Ref. |
|---|---|---|---|
| Wolfberry dietary fiber (goji berry) | 0.5–5% | Yogurt containing 2% (*w/v*) wolfberry dietary fiber had less syneresis, higher apparent viscosity, and increased hardness compared to control yogurt. | [235] |
| **Fruit or fruit ingredient and vegetable** | | | |
| Fruit purees (peach, apple, and pear) | 10 and 20% | Peach and apples were the most suitable fruits for probiotic yogurt. | [372] |
| Dragon fruit | 12% | The optimal formulation was 12% dragon fruit, 11% sugar, and 2% *L. plantarum*. Fermentation time was 19 h at 37 °C. | [373] |
| Isabel "Precoce" grape ingredients | Isabel grape preparation (20 g/100 mL) By-product flour (2 g/100 mL) | This goat milk yogurt had high *L. acidophilus* La-05 counts, distinct phenolic profile, higher antioxidant capacity, sensory acceptance, and consumer preference compared to control probiotic yogurt. | [374] |
| Orange sweet potato | 15 and 25% | Orange sweet potato purees incorporated into probiotic yogurt were accepted by consumers. | [375] |
| **Berry and nut** | | | |
| Gobdin (Dry white mulberry and walnut paste) | 0, 5, and 10% | Adding 5% gobdin to yogurt containing *L. acidophilus* resulted in an acceptable product. | [376] |
| **Juice (fruit or vegetable)** | | | |
| Pomegranate juice | 16% | Yogurt fortified with pomegranate juice and probiotics had desirable sensory properties during storage. | [377] |
| Carrot juice | 8, 16, 24, and 32% | There was increased color intensity, carrot flavor, creaminess, mouth coating, and chalkiness with increased carrot juice levels. | [378] |
| **Juice and flower** | | | |
| Juice from kiwifruit and jasmine flour | 20% kiwi fruit juice and 15% jasmine flower juice | The best formulation was 20% kiwi fruit juice, 15% jasmine flower juice, and 5% inoculum concentration. Fermentation time was 8 h at 40 °C. | [379] |
| **Spice and Oil** | | | |
| Spices (Cardamom, cinnamon, and nutmeg) | 0.5% (*v/w*) | Yogurts containing spices had good sensory properties with enhanced antioxidant activity. | [380] |
| Ginger and chamomile essential oil | 0.2 and 0.4% | Ginger and chamomile essential oils and *B. lactis* Bb12 addition enhanced yogurt properties. Incorporation of essential oil significantly decreased fermentation time. | [381] |
| Dill essential oil | 50 and 100 ppm | Yogurt containing 100 ppm dill essential oil received high sensory scores and maintained high viability of *B. bifidum* and *L. casei*. | [382] |
| Peppermint, Basil, and Zataria essential oils | 0.5% | Antioxidant potential was improved by addition of all three essential oils. Peppermint and basil yogurts had acceptable sensory properties, but zataria yogurt was not as acceptable. | [383] |
| **Bee products** | | | |
| Pine honey | 2, 4, and 6% | The 2% level was the preferred level during sensory evaluation. | [384] |
| Royal jelly | 2% (*w/v*) | Royal jelly incorporation Ssignificantly improved physicochemical, rheological, sensory, and microbiological properties (increased probiotic viability) compared to control probiotic yogurt. | [385] |

**Table 5.** *Cont.*

| Functional Ingredient Category (in Bold) and Ingredient | Concentration | Effect on Properties | Ref. |
|---|---|---|---|
| **Cyanobacterium** | | | |
| Spirulina (a biomass of cyanobacterium) | 1 g per liter of yogurt mix. | This yogurt was less acidic than the control yogurt on the 7th day, and there was higher growth of lactic acid bacteria in this yogurt than for the control yogurt on the 7th day. | [386] |

[1] RS2 is high amylose corn starch while RS3 is physically modified corn starch. [2] Inulin chain lengths were short (P95), medium (GR), and long (HP).

## 11. Safety of Yogurt

Although yogurt is generally a safe product because of the added starters added to, food poisoning outbreaks related to yogurt consumption have occurred as summarized by Aryana and Olson [387]. Furthermore, Aziz et al. [388] found pathogens (including *Streptococcus equinus*, *Escherichia fergusonii*, *Ralstonia pickettii*, and *Delftia tsuruhatensis*) in probiotic yogurt in Pakistan. Gram-negative psychrotrophic bacteria, yeast, and mold contamination must be avoided in yogurt and is more common in traditionally manufactured yogurt than in industrially manufactured yogurt [389]. Conversely, Montaseri et al. [390] found that probiotic yogurt can lower aflatoxin $M_1$ during storage.

The probiotics themselves must also be evaluated for their safety. Although lactobacilli are not generally pathogenic [391], Sims [392] has reported a *lactobacillus* (an oral strain of *L. casei* var. *rhamnosus*) that was lethal to mice and rats. *Bifidobacteria*, with the exception of the pathogenic *Bifidobacterium dentium*, have only rarely been found to be involved with certain dental and other infections [393]. Potential risk factors for consuming probiotics include systemic infections arising from bacterial translocation, antimicrobial resistance gene transfer to pathogenic bacteria, divergent immune stimulation in susceptible groups, and undesirable metabolic activities [394].

Although probiotics are frequently associated with positive health outcomes, there have been studies in certain populations in which subjects in the probiotic group had worse outcomes than subjects in the placebo group. Callaway et al. [395] found higher percent of cases of gestational diabetes mellitus (18.4% versus 12.3%, ($p = 0.10$)), higher oral glucose tolerance test results (79.3 mg/dL versus 77.5 mg/dL ($p = 0.049$)), and higher incidences of preeclampsia (9.2% versus 4.9% ($p = 0.09$)) in overweight and obese pregnant women who took capsules or sachets of *Lactobacillus rhamnosus* GG and *Bifidobacterium animalis* subsp. *lactis* BB-12. In the PROPATRIA (Probiotics in Pancreatitis Trial) study (a trial attempting to reduce infectious complications in patients with severe acute pancreatitis by supplying them with a multi-species probiotic preparation (freeze-dried Ecologic 641) delivered enterally), 16% (24 of 152) of patients died in the treatment (probiotic) group versus 6% (9 of 144) of the patients who died in the placebo group [396]. These authors thought that this high mortality rate for the treatment group was due to a lethal combination of pancreatic enzymes (mainly proteolytic) and probiotic therapy and resulting in the production of excessively high levels of lactic acid. They recommended that researchers immediately start probiotic therapy after the initial onset of disease, limit fermentable carbohydrate supply, prevent bacterial overgrowth of the microflora within the patient, and dramatically increase the probiotic bacterial dose [396]. Therefore, extra care must be taken when administering probiotics to critically ill patients or other susceptible populations.

## 12. Survival of Probiotics in Yogurt

Several studies have examined survival of probiotics in yogurt. Gilliland and Speck [397] reported that *L. acidophilus* does not survive well in yogurt during storage, probably because of hydrogen peroxide produced by *L. bulgaricus*. Ng et al. [398] concluded that the reason that counts of some strains of *L. acidophilus* decrease in the presence of *L. bulgaricus*

is not due to the low pH, as these strains of *L. acidophilus* can survive in a similar pH yogurt that was produced by glucono-delta-lactone. (Glucono-delta-lactone gradually releases gluconic acid at a comparable rate to acids produced by starter cultures.) Similar to Gilliland and Speck [397], Ng et al. [398] also suggested that this inhibition of some strains of *L. acidophilus* may have been caused by high concentrations of hydrogen peroxide produced by *L. bulgaricus*. Shah et al. [399] obtained five brands of commercial probiotic yogurt and determined how counts of *L. acidophilus* and *B. bifidum* changed over 5 weeks of refrigerated storage. Initial viable *L. acidophilus* counts were in the range of $10^7$ to $10^8$ cfu/g for three of the brands, around $10^6$ cfu/g for another brand, and around $10^4$ cfu/g for the final brand, and the three brands with the highest counts maintained their counts better than the two brands with lower initial counts. Initial viable *B. bifidum* counts were around $10^7$ cfu/g for two brands and between $10^3$ to $10^4$ cfu/g for the remaining three brands, and all of these counts decreased during storage, especially during the latter stages [399]. Mani-López et al. [400] reported that *L. acidophilus* maintained better viability than *L. reuteri* and *L. casei* during storage of yogurt and fermented milk containing *S. thermophilus*. Hekmat et al. [401] used *Lactobacillus reuteri* (now *Limosilactobacillus reuteri*) RC-14 and *Lactobacillus rhamnosus* (now *Lacticaseibacillus rhamnosus*) GR-1 in the preparation of yogurt and found that *L. rhamnosus* GR-1 was surviving better than *L. reuteri* RC-14. For *Limosilactobacillus mucosae* CNPC007 incorporated into goat milk Greek-style yogurt, de Morais et al. [30] reported that counts decreased from 9.53 log cfu/g at 1 day of storage to 8.96 log cfu/g at day 28.

Some factors may affect survival of probiotics in yogurt during storage. Kailasapathy et al. [402] manufactured fruit yogurts containing *L. acidophilus* and *B. animalis* ssp. *lactis* and found that the fruit preparation usually did not decrease viability of these probiotics during storage compared to their plain yogurt controls. Ferdousi et al. [403] found faster declines of viable counts of various types of probiotics in yogurt stored at 20 °C compared to 5 °C, and *L. rhamnosus* HN001 maintained better viability than *B. animalis* subsp. *lactis* BB-12).

Viability of probiotics in frozen yogurt has been reported in the literature. Hekmat and McMahon [404] manufactured frozen yogurt by fermenting an ice cream mix with *L. acidophilus* and *B. bifidum* before freezing and found decreases in counts from $1.5 \times 10^8$ cfu/mL to $4 \times 10^6$ cfu/mL for *L. acidophilus* and from $2.5 \times 10^8$ cfu/mL to $1 \times 10^7$ cfu/mL for *B. bifidum* during 17 weeks of frozen storage at −29 °C. Davidson et al. [405] manufactured frozen yogurt containing *L. acidophilus* and *B. longum* and found little to no change in culture survival during storage at −20 °C for 11 weeks. Atallah et al. [406] manufactured frozen yogurt using either sucrose or sugar replacements as the sweetener and found decreases in *B. bifidum* counts from about 7.6 log cfu per gram at 1 day of storage to about 6.2 to 6.3 log cfu per gram at 60 days of frozen storage.

Many steps can be taken to improve viability of probiotics in yogurt or frozen yogurt during storage. Although *L. acidophilus* counts in yogurt can be raised by increasing its inoculation level during manufacture to a certain extent, *L. acidophilus* inoculation levels that are excessively high lowered the counts during storage, resulting in a lower quality of yogurt including reduced apparent viscosity and sensory scores but increased syneresis compared to yogurts produced with lower inoculation levels of *L. acidophilus* [35]. A method has been patented [407] for enhancing growth and viability of *L. acidophilus* in yogurt by inoculating this microorganism into a base that includes heat treated and cooled milk and possibly other ingredients including fiber) and then growing this microorganism during incubation. *S. thermophilus* and *L. bulgaricus* are then inoculated into this yogurt mix containing the *L. acidophilus* culture for further incubation to form a yogurt with significantly higher counts of *L. acidophilus* but typical counts of *S. thermophilus* and *L. bulgaricus* [407]. Incorporating increasing concentrations of ascorbic acid of up to 250 mg/kg of probiotic yogurt led to a slower decrease in *L. acidophilus* counts during storage [408]. The viability of bifidobacteria was improved by the addition of cysteine, whey protein concentrate, acid casein hydrolysates, and tryptone, but not by the addition of dried

whey [409]. Costa et al. [410] reported that incorporating oligofructose or polydextrose to a probiotic yogurt containing *L. casei* improved probiotic survival and texture but lowered flavor acceptance. Sarwar et al. [71] manufactured synbiotic yogurt incorporating 0.5% *Saccharomyces boulardii* CNCM I-745 probiotic yeast and up to 2% inulin and found a slower decrease in viability of *S. boulardii* during 4 weeks of refrigerated storage with yogurts containing increasing inulin concentrations. Muzammil et al. [411] manufactured frozen yogurt supplemented with inulin and glycerol and found smaller decreases in viability of *L. acidophilus* and *B. lactis* in frozen yogurt when supplemented with up to 6% inulin or 4% glycerol after 12 weeks of storage. Although presence of oxygen (oxidative stress) is undesirable in yogurt (toxic to some cells, leads to production of hydrogen peroxide by certain strains, and production of free radicals from food component oxidation), addition of glucose oxidase for oxygen removal maintained probiotic culture viability and lactic and acetic acid levels but increased diacetyl, acetaldehyde, conjugated linoleic acid, and polyunsaturated fatty acid levels in yogurt in the study of Batista et al. [412].

Microencapsulation is a process in which at least one potentially sensitive substance becomes entrapped by a coating material for its protection and has recently been reviewed by Gullo and Zotta [413]. Pour et al. [414] prepared yogurt incorporating either free or encapsulated probiotics (*L. rhamnosus* and *L. plantarum*) and found increased survival with probiotics encapsulated in a multi-layer emulsion (decrease of 7.59–7.65 log cfu/mL at 1 day of storage to 7.45–7.55 log cfu/mL at 21 days of storage) versus free probiotics (7.59–7.71 log cfu/mL to 6.82–6.93 log cfu/mL for the same times). Ajlouni et al. [415] encapsulated *L. acidophilus* LA-5 and *B. lactis* BB-12 and added these to yogurt either before or after yogurt fermentation. Although the encapsulated probiotic count decreased in yogurt stored for 21 days under refrigeration regardless if added before or after fermentation, the probiotic count increased after 24 h of in-vitro colonic fermentation, even after 21 days of storage. These results indicated that the encapsulated probiotics would be more bioaccessible in the colon. Incorporation of encapsulated probiotics before fermentation resulted in higher counts than incorporation after fermentation [415]. Dimitrellou et al. [41] freeze-dried *Lactobacillus casei* ATCC 393 on casein and apple pieces to be used as an adjunct culture for producing yogurt and found a lower pH, higher titratable acidity, less syneresis, altered concentration of key volatile compounds, and improved sensory properties compared to yogurts produced traditionally. They also reported detection of greater than $10^7$ log cfu per gram of this probiotic after 28 days of storage.

Ultrasound and packaging can be used to improve probiotic viability. The fermentation time during yogurt manufacturing was reduced by 30 min when applying sonication after inoculation [416]. Use of glass bottles and thicker plastic packaging can reduce oxygen permeability into yogurt and better maintain probiotic survival [417].

Probiotics must survive not only during storage throughout its shelf-life, it must also survive within the body for it to confer health benefits to its host. These probiotics must survive the acidic conditions within the stomach and bile salts in the intestinal tract. In vitro tests performed in the lab can be performed to determine if the probiotic can survive in broth adjusted to a low pH (typically pH 2) in the acid tolerance test and in broth to which a bile salt (oxgall) has been added in the bile tolerance test. Acid and bile tolerance of probiotics from dairy and nondairy products was recently reviewed by Ayyash et al. [418].

## 13. Properties of Probiotic Yogurt

Some research projects have investigated effects of probiotic incorporation on physico-chemical and sensory properties of probiotic yogurts. Cui et al. [419] produced cow milk yogurt with yogurt starter cultures alone and with yogurt starter cultures combined with probiotics (*Bifidobacterium animalis* subsp. *lactis* BB-12, *L. acidophilus* La-5, and *L. rhamnosus* GG) and found probiotic supplementation decreased time to reach pH 4.5 and pH obtained during storage but increased initial firmness. However, syneresis, color, and microstructure were not affected by probiotic supplementation. Soni et al. [420] prepared yogurt by incorporating individual probiotics (*L. acidophilus*, *L. casei*, *L. plantarum*, and *B. bifidum*)

and certain combinations of these probiotics, and found that the incorporated probiotics affected various nutritional, physicochemical, organoleptic, and probiotic properties. Additionally, these authors found that using a combination of probiotics resulted in improved texture and often better probiotic potential. He et al. [421] reported that a higher viscosity resulting from probiotic addition is due to increased total solids content and increased packing of the three-dimensional casein cluster network arising from exopolysaccharide production. Similarly, de Morais et al. [30] reported a higher viscosity for their probiotic goat milk Greek-style yogurt than for their control yogurt, probably arising from increased exopolysaccharide production by their *Limosilactobacillus mucosae* adjunct culture.

Mixed results have been reported for the effects of probiotic incorporation into yogurt on sensory properties. Some studies found that addition of probiotics to fermented milk or yogurt does not significantly affect sensory properties [400,422,423]. However, Hussain et al. [424] purchased probiotic and natural yogurt in the UK and found that the probiotic yogurt was organoleptically favored over the natural yogurt. Likewise, the probiotic goat milk Greek-style yogurt in the de Morais et al. [30] study received higher sensory scores for color, flavor, texture (at 28 days), and overall acceptance than their control yogurt.

## 14. Greek Yogurt Acid Whey

Production of Greek yogurt leaves acid whey as a by-product. Although this Greek yogurt acid whey is commonly spread on fields for use as a fertilizer or fed to livestock, it does have potential uses in foods. Smith et al. [425] neutralized this Greek yogurt acid whey and claimed that it can be incorporated into bakery, beverage, snack, confectionery, soup, dry meal, dairy, and cereal products. Food uses for Greek yogurt acid whey can include utilization in ranch dressing [426], in pancake and pizza crust [427], and milk protein concentrate-based extruded snack product [428]. Rivera Flores et al. [429] prepared a beverage by fermenting Greek-style yogurt acid whey using pure cultures of *Saccharomyces cerevisiae*, *Kluyveromyces marxianus*, *Brettanomyces claussenii*, or *Brettanomyces bruxellensis* and using a yeast nitrogen base supplemented with lactose, glucose, or a 1:1 mixture of glucose and galactose under aerobic conditions. For the glucose and galactose mixture for *B. clausenii*, all of the glucose was consumed with acetic acid production, but galactose was not utilized, conferring this beverage with prebiotic properties. Dufrene et al. [430] manufactured a pineapple-flavored probiotic acid whey drink incorporating *L. acidophilus* and found some survival of *L. acidophilus* after 4 weeks of storage.

## 15. Conclusions

A wide range of probiotic strains can be added to yogurt and yogurt-like products. Many different innovations and unique selling points can be found for probiotic yogurt already on the market. There are many different forms of probiotic yogurt (spoonable, drinkable, concentrated (Greek), dried, low lactose, shelf-stable, frozen), and probiotic yogurt may be used as an ingredient, including as a coating or as a snack, in many other types of foods. Many research papers have described useful functional ingredients that have been added to probiotic yogurt. It is important to maintain viability of these probiotics during their shelf-life and within the body for the consumer to obtain the health benefits, and many factors and technologies can be used to improve their probiotic counts and their shelf-life. As with any type of food product, ensuring food safety is critical, especially for critically ill patients and other susceptible populations. Even the acid whey by-product from Greek yogurt manufacturing can be used as an ingredient in other food products or incorporated with probiotics to form a probiotic by-product beverage. The wide variety of available probiotics, methods for improving probiotic viability, and forms and uses of probiotic yogurt present many exciting opportunities for new product development to improve sales and consumer health.

**Author Contributions:** Writing—original draft preparation, D.W.O.; Writing—review and editing, D.W.O. and K.J.A. All authors have read and agreed to the published version of the manuscript.

**Funding:** This research received no external funding.

**Institutional Review Board Statement:** Not applicable.

**Informed Consent Statement:** Not applicable.

**Conflicts of Interest:** The authors declare no conflict of interest.

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
