# Peer review of "Probiotic Incorporation into Yogurt and Various Novel Yogurt-Based Products"

_applsci, doi:10.3390/app122412607_

Round 1

Reviewer 1 Report

The manuscript reviews about the probiotic incorporation into yogurt and various novel yogurt-based products. Here my comment regarding the manuscripts.

1.       The review touches many topics on the incorporation of probiotic in yogurt. However, each topic is not thoroughly discussed. The advancement of the techniques, newly discovered benefit of probiotics in yogurt, new probiotic strains etc. should be focused more rather than lengthy discussing the information that are discovered a long time ago.

2.       Many topics in this manuscript have already reviewed elsewhere. The authors used these reviews in the manuscript without significantly adding new additional information.

3.       The subtopics are not properly organized. For example, the subtopic of “Types of probiotics” can be further divided into other subsubtopics to discussed the individual probiotics.

4.       The author lengthy discussed about prebiotics, synbiotics, paraprobiotics, postbiotics, and psychobiotics. These topics should be just briefly touch and focusing more on probiotic.

5.       The title states that the manuscript is about yogurt and various novel yogurt-based products. However, the yogurt-based product that being discussed is just acid whey. The discussion on the acid whey is also too brief.

Overall, the review should more specific and deeply discussed on the new discovery in recent years rather than briefly touching every aspect on the topics. In my opinion, the review is not ready for publication since the new discovery on the inclusion of probiotics in yogurt is not thoroughly discussed.

Author Response

Dear Reviewer 1,

  1. We wrote this review paper for both researchers and for dairy industry personnel, who may or may not have a strong science background. For the benefit of the readers who do not have a strong scientific background, we wanted to introduce many of the basic concepts without going into too much detail.  Some techniques such as microencapsulation and preparation of paraprobiotics have been covered.  Many recent publications about specific health benefits provided by probiotics have been added.  Also, paragraphs about next generation probiotics and precision probiotics have been added.

  1. Since this paper was getting long, we wanted to refer readers to more detailed information elsewhere if they wanted to learn more about a particular topic instead of making this review paper even longer.

  1. We believe that this revision is better organized than the original version. For example, Tables 1 and 5 have been subdivided into different genera and different categories of useful functional ingredients.

  1. We believe that the topics of prebiotics, synbiotics, paraprobiotics, postbiotics, and psychobiotics are also important since they also provide health benefits.

  1. When we mentioned various novel yogurt-based products, we meant the sections entitled “Various Forms of Yogurt” and “Use of Probiotic Yogurt as an Ingredient”. The section dealing with acid whey has been expanded.

Overall,     Much new information including many new references has been added to this revision, and it is better organized than the first revision.  Many topics are now discussed in more detail.  Three new tables (a list of probiotic bacteria and yeast found in yogurt, a list of probiotics and prebiotics found in synbiotic yogurt, and a list of useful functional ingredients found in probiotic yogurt) have been added, and one table (the health benefits table) has been expanded.  We hope that this review paper is now much stronger than the original version and now ready for publication.

Thank you. 

Reviewer 2 Report

This is an interesting and nicely written review of various probiotic yogurt and yogurt-based products. In addition to probiotics, the terms prebiotics, synbiotics, paraprobiotics, postbiotics, and psychobiotics are clearly defined, contributing to a better understanding of their differences.

My minor comments are related to section 4. Table 1 is less informative due to a lack of overall clinical effects for mentioned health conditions. However, the authors explained this imperfection by the complexity of evaluating the strength of the evidence, so they did not consider it.

The sentence in lines 133,134 seems to be incomplete. Check this!

Add punctuation in line 461.

The principal definition of additives is "Food additives are substances added intentionally to foodstuffs to perform certain technological functions, for example, to colour, to sweeten or to help preserve foods.”

I would suggest renaming section 10 from “Additives” to “Useful functional ingredients in probiotic yogurt,” which is more in the context of this section.

References are not adequately listed. Follow the Instruction for Authors.

This review provides valuable details regarding different probiotic strains in various forms of probiotic yogurt.

The paper complies with the field of this journal.

Author Response

Dear Reviewer 2,

REVIEWER COMMENT “This is an interesting and nicely written review of various probiotic yogurt and yogurt-based products.  In addition to probiotics, the terms prebiotics, synbiotics, paraprobiotics, postbiotics, and psychobiotics are clearly defined, contributing to a better understanding of their differences.

My minor comments are related to section 4.  Table 1 is less informative due to a lack of overall clinical effects for mentioned health conditions.  However, the authors explained this imperfection by the complexity of evaluating the strength of the evidence, so they did not consider it”.

RESPONSE:  Although it would have been good to discuss the clinical effects of the health benefits, it would be difficult to do so.  Since the main emphasis of this paper is about probiotic yogurt, we believe that a fairly extensive list, consisting of about 100 references in one table, is adequate to give the reader an idea of the health benefits provided by probiotics.

REVIEWER COMMENT  “The sentence in lines 133, 134 seems to be incomplete.  Check this!”

RESPONSE:  Lines 133 and 134 are a heading and not meant to be a complete sentence.

REVIEWER COMMENT  Add punctuation in line 461.

RESPONSE:  This sentence was deleted, and the information within this paragraph was incorporated into Table 5.

REVIEWER COMMENT The principal definition of additives is “Food additives are substances added intentionally to foodstuffs to perform certain technological functions, for example, to colour, to sweeten or to help preserve foods.

I would suggest renaming section 10 from “Additives” to “Useful functional ingredients in probiotic yogurt” which is more in the context of this section.

RESPONSE:  This change has been made in the abstract, the paragraph describing the useful functional ingredients, the conclusion, and Table 5 (a new table in this revision).

REVIEWER COMMENT References are not adequately listed.  Follow the Instructions for Authors.

RESPONSE:  The references were re-formatted to MDPI style, and cited as numbers instead of names in this revised version.

REVIEWER COMMENT This review provides valuable details regarding different probiotic strains in various forms of probiotic yogurt.

RESPONSE:  Thank you for your comment.

REVIEWER COMMENT  The paper complies with the field of this journal.

RESPONSE:  Thank you.
